# Activity of the *C. elegans* egg-laying behavior circuit is controlled by competing activation and feedback inhibition

Kevin M Collins[1,2]*, Addys Bode[1], Robert W Fernandez[2], Jessica E Tanis[2†], Jacob C Brewer[2], Matthew S Creamer[3‡], Michael R Koelle[2,3]

[1]Department of Biology, University of Miami, Coral Gables, United States; [2]Department of Molecular Biophysics and Biochemistry, Yale University, New Haven, United States; [3]Interdepartmental Neuroscience Program, Yale University, New Haven, United States

*For correspondence: kcollins@bio.miami.edu

Present address: †Department of Biological Sciences, University of Delaware, Newark, United States; ‡Department of Molecular, Cellular, and Developmental Biology, Yale University, New Haven, United States

Competing interests: The authors declare that no competing interests exist.

**Abstract** Like many behaviors, *Caenorhabditis elegans* egg laying alternates between inactive and active states. To understand how the underlying neural circuit turns the behavior on and off, we optically recorded circuit activity in behaving animals while manipulating circuit function using mutations, optogenetics, and drugs. In the active state, the circuit shows rhythmic activity phased with the body bends of locomotion. The serotonergic HSN command neurons initiate the active state, but accumulation of unlaid eggs also promotes the active state independent of the HSNs. The cholinergic VC motor neurons slow locomotion during egg-laying muscle contraction and egg release. The uv1 neuroendocrine cells mechanically sense passage of eggs through the vulva and release tyramine to inhibit egg laying, in part via the LGC-55 tyramine-gated Cl⁻ channel on the HSNs. Our results identify discrete signals that entrain or detach the circuit from the locomotion central pattern generator to produce active and inactive states.

## Introduction

Neural circuits are the functional units underlying all thoughts and behaviors, but there is as yet no neural circuit in any organism for which we know with precision how the many signaling events among its cells together generate its activity. This would require, among other things, understanding how neuromodulators shape activity at chemical and electrical synapses to alter the excitability of specific cells in the circuit to generate regulated, dynamic patterns of circuit responses and a coherent behavioral output. One approach to this problem is to analyze small neural circuits typical of invertebrate model organisms, so that both the simplicity of the circuits and the powerful experimental approaches uniquely possible within these systems can provide a penetrating analysis (*Marder, 2012*).

*C. elegans* egg-laying behavior is controlled by a small circuit that offers many experimental advantages for study (*Schafer, 2006*). This circuit, diagrammed in *Figure 1A*, consists of two serotonergic Hermaphrodite Specific Neurons (HSNs) and six cholinergic Ventral C neurons (VCs), each of which synapse onto a set of vulval muscles whose contraction expels eggs. Four neuroendocrine uv1 cells also regulate egg laying (*Jose et al., 2007*). Despite its anatomical simplicity, the egg-laying circuit produces a regulated, rhythmic behavior that alternates between quiescent periods of about 20 min during which no egg laying occurs, and active states lasting a few minutes during which ~5 eggs are laid (*Waggoner et al., 1998*). Active states appear to result when the HSNs release serotonin

**eLife digest** It has been said that if the human brain were so simple that we could understand it, we would be so simple that we couldn't. This quote neatly captures the challenge of working out how 80 billion neurons collectively generate our thoughts and behavior. Fortunately, the nervous system is also organized into simpler units called circuits. Each consists of a relatively small number of neurons, which communicate with one another to control as little as a single behavior. These circuits should in principle be simple enough for us to understand, particularly if we study them in nervous systems less complex than our own.

Despite this, there is currently not a single circuit in any organism in which we can explain how communication between individual neurons generates behavior. Collins et al. therefore set out to characterize a simple neural circuit in one of the simplest model organisms: the egg-laying circuit of the worm *C. elegans*.

Using mutations, drugs and molecular genetic techniques, Collins et al. systematically altered the activity and signaling of each of the neurons within the egg-laying circuit. The experiments revealed that cells called command neurons trigger egg laying by producing signals that switch on the rest of the circuit. Once activated, the circuit is able to respond to waves of activity from a second circuit – called the central pattern generator – that also controls the worm's movement. Finally, laying an egg activates a third set of neurons, which release a signal that returns the circuit to its inactive state.

The use of distinct signals and neurons to activate the circuit, to coordinate its ongoing activity, and to inactivate the circuit when its task is complete also applies to many other neural circuits. Now that these signals have been identified in one circuit, it should be possible to build on these findings to better understand how others work.

that signals through G protein coupled receptors on the vulval muscles to increase their excitability (*Waggoner et al., 1998*; *Shyn et al., 2003*; *Hapiak et al., 2009*; *Emtage et al., 2012*). Strong regulation by sensory stimuli is superimposed on the pattern of alternating behavioral states. For example, carbon dioxide regulates neuropeptide release from head sensory neurons that signal through receptors on the HSNs to inhibit egg laying (*Ringstad and Horvitz, 2008*; *Hallem et al., 2011*). Worms also halt egg laying in the absence of food and restart the behavior when re-fed (*Daniels et al., 2000*; *Dong et al., 2000*).

*C. elegans* egg laying has been studied for decades, and dozens of genes have been identified by mutations that either reduce (*Desai et al., 1988*) or increase (*Bany et al., 2003*) egg laying. Some of the identified genes encode ion channels that regulate cell and synaptic electrical excitability (*Elkes et al., 1997*; *Johnstone et al., 1997*; *Lee et al., 1997*; *Weinshenker et al., 1999*; *Jospin et al., 2002*; *Jose et al., 2007*; *Collins and Koelle, 2013*). Other identified genes encode components of G protein signaling pathways that act in specific cells of the circuit (*Brundage et al., 1996*; *Koelle and Horvitz, 1996*; *Hajdu-Cronin et al., 1999*; *Miller et al., 1999*; *Williams et al., 2007*; *Porter et al., 2010*). These results suggest neuromodulators, including serotonin, signal through G proteins to regulate the excitability of specific cells in the circuit to control circuit activity and egg laying.

The ability to leverage egg-laying mutants to understand the molecular mechanisms that initiate, sustain, and terminate the active state of egg laying has been limited by the tools available to analyze activity in the cells of the egg-laying circuit. $Ca^{2+}$ activity in cells in the circuit was initially optically recorded in immobilized animals, which have limited ability to engage in egg-laying behavior (*Shyn et al., 2003*; *Zhang et al., 2008*). More recently we developed methods to optically record $Ca^{2+}$ activity of the vulval muscles in moving animals as they engage in normal egg-laying behavior. Using ratiometric imaging to correct for movement and focus artifacts, and recording at 20 frames/sec for periods of up to one hour, we were able to measure activity of the vulval muscles as animals cycled through active and inactive states (*Collins and Koelle, 2013*; *Li et al., 2013*). Our recordings showed that the vulval muscles are excited rhythmically in phase with the body bends of animal locomotion, and we proposed this was due to signaling from the cholinergic motor neurons, including the VCs, that are rhythmically active during locomotion. We found that a conserved ERG $K^+$ channel

depresses response of the vulval muscles to excitation, thus maintaining the inactive behavior state, and we proposed that serotonin signaling onto vulval muscle receptors enhances vulval muscle excitability to allow the strong muscle contractions and egg release seen in the active state. We have now generated tools to carry out $Ca^{2+}$ recordings in moving animals for each neuron type in the egg-laying circuit. Combining these recordings with precise manipulations of the circuit using mutations, optogenetics, and drugs allows us to deduce how specific signals from these cells drive specific features of circuit activity and behavior.

## Results

### Distinct patterns of activity in the egg-laying circuit accompany the active and inactive behavior states

We carried out long-term $Ca^{2+}$ recordings of HSN and VC neuron activity in behaving animals using methods we developed previously for the vulval muscles (*Collins and Koelle, 2013*). We generated transgenic strains of *C. elegans* co-expressing the fluorescent $Ca^{2+}$ reporter GCaMP5 and the $Ca^{2+}$-independent fluorescent protein mCherry specifically in the HSN or VC neurons, and used ratiometric imaging to record cell activity and egg-laying behavior as animals moved through active and inactive egg-laying states (*Figure 1*). Because of the large size of the egg-laying synapse, we were able to clearly observe $Ca^{2+}$ transients localized to the presynaptic terminus in HSN (*Figure 1B*; *Video 1*) and VC (*Figure 1C*; *Video 2*). For comparison, we also carried out $Ca^{2+}$ recordings of vulval muscles, and as in our earlier studies of these muscles, were able to clearly distinguish the weak twitching contractions (*Figure 1D*; *Video 3*) from the strong contractions that drove egg release (*Figure 1E*).

Through long-term $Ca^{2+}$ imaging, we observed a striking increase in rhythmic activity in the cells of the circuit that began prior to the first egg-laying event and persisted beyond the last egg-laying event of each active state (*Figure 1F*). We defined the active state as one minute before the first egg-laying event and one minute after the last egg-laying event.

We found that HSNs displayed long (~4 s at half-maximum amplitude) $Ca^{2+}$ transients that were largely consistent in waveform and amplitude, but that varied in occurring either as single events or trains (*Figure 1F*). There was also significant HSN activity outside of the active state, but HSN transients were more frequent during the active state (*Figure 1G*). Active state HSN transients typically occurred in bursts, within which HSN transients occurred every ~20 s (*Figure 1G*). Only 11% of HSN transients were accompanied by an egg release (*Figure 1H*), but almost every egg-laying event occurred during an HSN transient (*Figure 1F*; out of 49 total egg-laying events observed, 48 occurred within an HSN transient).

VC transients were shorter than HSN transients (~2 s at half-max) and variable in amplitude. We rarely observed VC transients during the inactive state, but we found a clear induction of transients during the active state (*Video 2*; *Figure 1F,G*), and these transients occurred in bursts around egg-laying events with an inter-transient interval of ~10 s, about twice as frequent as seen for the HSNs. Although $Ca^{2+}$ transients were most intense in the VC presynaptic varicosities and axonal processes, they were also visible in all six VC cell bodies, especially those of the VC4 and VC5 neurons that most closely flank the vulva (*Figure 1C* and *Video 2*). Vulval muscle twitching and egg-laying contractions caused significant mechanical deformation of the VC neurons. 34% of VC transients were accompanied by an egg release, a larger fraction than seen for HSN and vulval muscle $Ca^{2+}$ transients (p<0.0001; Fisher's exact test). Every egg-release event was accompanied by a VC transient. There was no correlation seen between the magnitudes of VC or HSN transients that did or did not coincide with egg release, while the magnitude of muscle $Ca^{2+}$ transients that resulted in egg laying were ~four fold larger than the muscle twitch transients that did not release eggs (*Figure 1H*).

The close apposition of HSN, VC, and vulval muscle synaptic regions prevented us from simultaneously imaging more than one cell type, but we were able to compare timing of activity in the different cells using the moment of egg release as an objective landmark. HSN $Ca^{2+}$ transients peaked ~2 s prior to egg release (*Figure 1I*) and decayed slowly. Shorter VC $Ca^{2+}$ transients followed HSN activity, peaking ~0.1 s before egg release. Vulval muscle $Ca^{2+}$ transients peaked with egg laying. We also tracked vulval muscle contraction by measuring the size and fluorescence intensity of the mCherry labeled muscles, and saw that muscle relaxation and decay of vulval muscle $Ca^{2+}$ immediately followed egg release (*Figure 1I*).

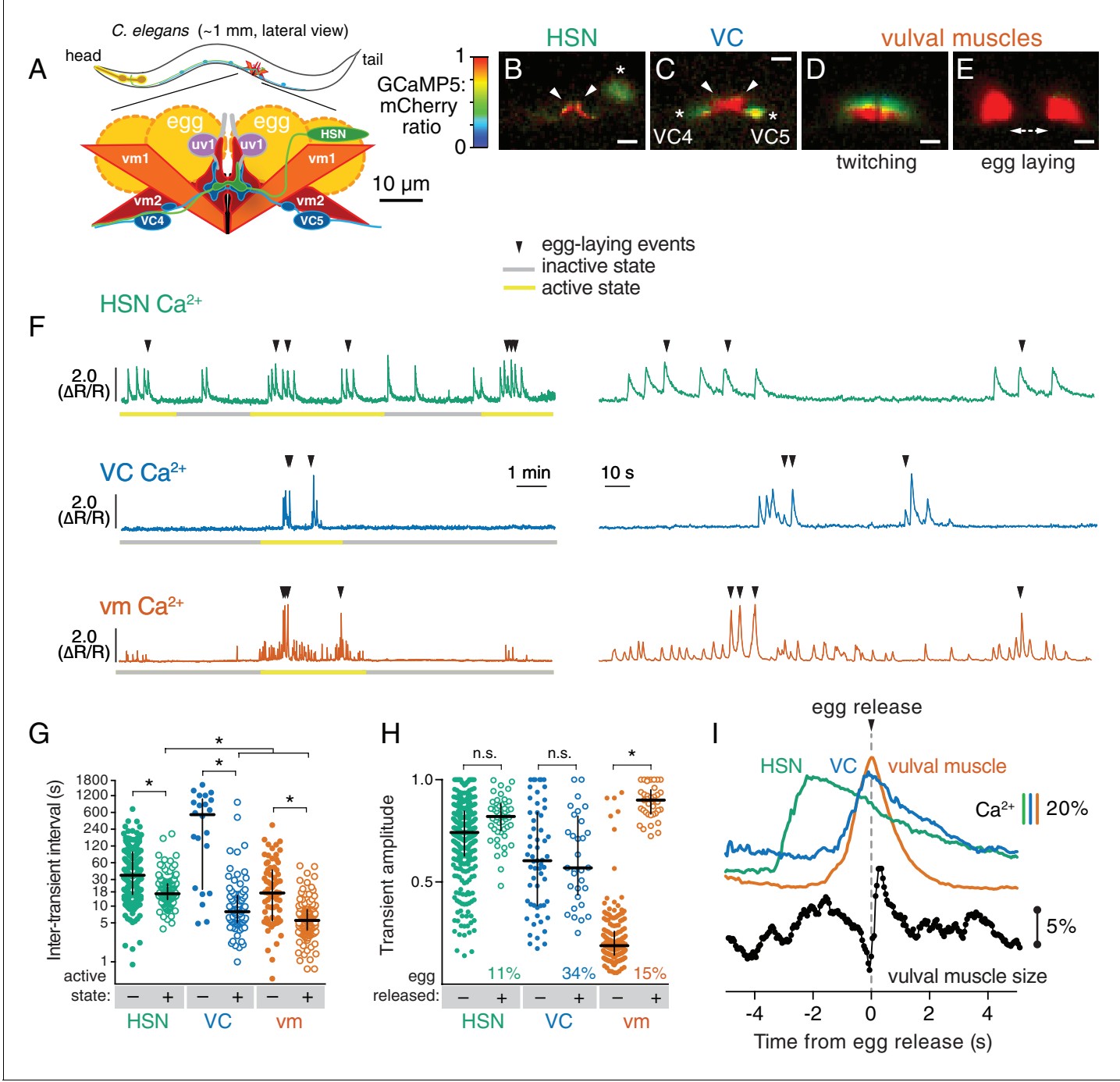

**Figure 1.** Cell-specific reporters of activity in the *C.elegans* egg-laying behavior circuit. (**A**) Schematic of the circuit. HSN (green) and VC (blue) motor neurons synapse onto the vm2 muscle postsynaptic termini (center of schematic). The uv1 neuroendocrine cells (pink) extend processes (grey) along the vulval slit and vm2 postsynaptic terminus. (**B–E**) Individual video frames of the GCaMP5:mCherry fluorescence ratio showing active state Ca²⁺ transients in HSNs (**B**), VCs (**C**), and vulval muscles during twitching (**D**) and egg-laying behaviors (**E**). Arrowheads, HSN and VC presynaptic termini; asterisks, cell bodies; scale bar, 10 µm. (**F**) 30 min recordings of HSN, VC, and vulval muscle activity (left panel), showing distinct active (yellow) and inactive (grey) egg-laying behavior states, with expanded timescale of one active state at right. Arrowheads show egg-laying events. (**G**) Scatter plots and median HSN, VC, and vulval muscle (vm) inter-transient intervals during egg-laying inactive (–, filled circles) and active (+, open circles) states. Asterisks indicate significant differences (p<0.0001). (**H**) Relationship between Ca²⁺ transient amplitude and egg release. Scatter plots and medians of normalized amplitude with (+; open circles) and without (–; closed circles) egg release. Also shown is the percent of total transients that accompanied egg release. (**I**) Timing of HSN, VC, and vulval muscle Ca²⁺ transients and egg release. Shown at top is a curve of the median of Ca²⁺ from HSN (green), VC (blue), and vulval muscles (orange) from normalized ΔR/R traces (with the peak Ca²⁺ set to 100%) synchronized to the moment of egg release (0 s, arrowhead

*Figure 1 continued on next page*

*Figure 1 continued*

and dotted line). Bars indicate 20% change in median GCaMP5/mCherry ratio. The timing of the HSN $Ca^{2+}$ peak is significantly different from that of the VCs and vulval muscles ($p<0.0001$). Shown at bottom is a trace of median vulval muscle size. Bar shows a 5% change in median object size based on mCherry fluorescence.

## HSN, VC, and vulval muscle activity are rhythmic and phased with animal locomotion

Previous work showed that timing of egg-laying events was not rhythmic but followed a Poisson distribution (*Waggoner et al., 1998*). However, we found that HSN, VC, and vulval muscles showed rhythmic activity during the egg-laying active state. We calculated the power spectra of HSN, VC, and vulval muscle $Ca^{2+}$ traces, and we observed three activity rhythms in the egg-laying circuit during the active state (*Figure 2A–E*). Rhythms in HSN and VC were similar, peaking at ~50 mHz with smaller peaks observed at ~100 and~140 mHz (*Figure 2B and C*). The 50 mHz rhythm was also clearly evident in the vulval muscles (*Figure 2D*), but the 100 mHz peak was significantly stronger (*Figure 2E*). These results show that despite egg-laying behavior being aperiodic, circuit activity underlying that behavior is rhythmic.

What is the source of circuit rhythmicity? *C. elegans* moves with sinusoidal body bends consisting of rhythmic waves of body-wall muscle contraction driven by the Ventral A and B (VA/VB) ventral nerve cord motor neurons. We previously observed that vulval muscle $Ca^{2+}$ transients tend to occur when the vulva is at a particular phase of the locomotion body bend (*Collins and Koelle, 2013*), suggesting rhythmic activity in the locomotor circuit may be the source of rhythmicity in the egg-laying circuit. Using our video recordings of $Ca^{2+}$ activity in moving animals, we examined whether HSN and VC activity, like vulval muscle activity, are phased with locomotion. Thus we determined timing of the peaks of HSN, VC, and vulval muscle $Ca^{2+}$ transients relative to when the vulva passed through the previous and subsequent most contracted or relaxed state of a body bend (*Figure 2— figure supplement 1*, see Materials and Methods). The results were plotted on histograms with ventral contraction at 0 and 360°, ventral relaxation at 180°, and intermediate phases of body bends at the angles in between (*Figure 2F–I*). We found

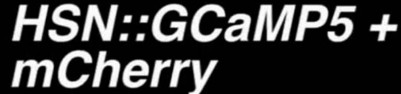

**Video 1.** Ratio recording of HSN $Ca^{2+}$ transients during the egg-laying active state. High $Ca^{2+}$ is indicated in red while low calcium is in blue. Large panel is an expanded view showing the freely-moving animal that has been contrast enhanced to make the worm and its laid eggs visible. Inset is cropped to a small area containing the HSN, and stabilized to remove movement. Text labels indicate when $Ca^{2+}$ transients and egg release occur.

**Video 2.** Ratio recording of VC $Ca^{2+}$ transients during the egg-laying active state. High $Ca^{2+}$ is indicated in red while low calcium is in blue. Contrast is enhanced to make the worm visible, although the laid eggs are not easily visible. Text labels indicate when vulval muscle (vm) twitches or egg-laying contractions occur, which result in small or large displacements of the VC4 and VC5 cell bodies, respectively.

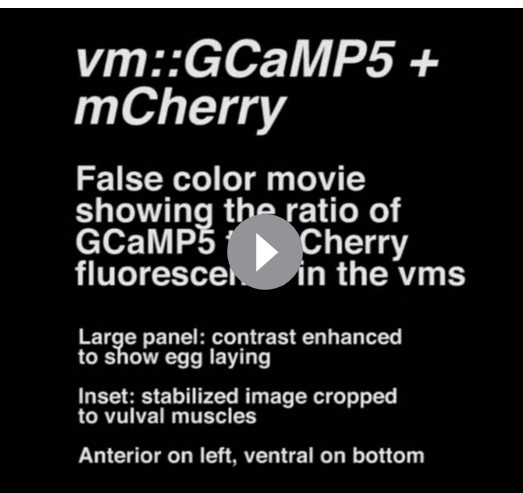

**Video 3.** Ratio recording of vulval muscle $Ca^{2+}$ transients during the egg-laying active state. High $Ca^{2+}$ is indicated in red while low calcium is in blue. Large panel is an expanded view showing the freely-moving animal that has been contrast enhanced to make the worm and its laid eggs visible. Inset is cropped to a small area containing the vulval muscles, and stabilized to remove movement. Text labels indicate when $Ca^{2+}$ transients and egg release occur.

that the HSN, VC, and vulval muscle $Ca^{2+}$ transients occurred when the vulva was at particular phases of the locomotor body bend. HSN transients just prior to egg release reached their maximum at ~180°, when the vulva and adjacent body wall muscles were at their most relaxed state (*Figure 2G*). VC and vulval muscle transients that accompanied egg laying occurred later, starting at ~225° (*Figure 2H and I*). HSN transients that did not result in egg laying showed no clear evidence of phasing (*Figure 2G*). In contrast, VC transients and vulval muscle twitch transients that did not lead to egg release were still phased, but delayed by ~45° toward the most ventral contracted state. These results confirm our previous observations that vulval muscle twitching and egg-laying activity are phased with locomotion (*Collins and Koelle, 2013*). These results extend that analysis to show that HSN, VC, and vulval muscle activity during egg laying events are rhythmic, phased with locomotion, and occur earlier in the body bend cycle, when the adjacent body wall muscles are in a more relaxed state.

## Optogenetic activation of HSNs initiates the egg-laying active state

To directly test how neurotransmitter signaling from the HSNs regulates egg-laying circuit activity, we used the *egl-6* promoter to express Channelrhodopsin-2 in the HSNs (*Emtage et al., 2012*), allowing us to drive neurotransmitter release specifically from the HSNs with blue light. We repeated the observation of *Emtage et al. (2012)* that optogenetic activation of HSNs stimulates egg-laying behavior (*Video 4*), but we activated the HSNs while simultaneously recording behavior and $Ca^{2+}$ activity in the VCs (*Figure 3A*) or vulval muscles (*Figure 3B*). We found that activation of HSNs resulted in circuit activity reminiscent of a spontaneous active state, including rhythmic $Ca^{2+}$ activity of both VCs and vulval muscles, and egg-laying events that accompanied a subset of these $Ca^{2+}$ transients. Some differences from a spontaneous active state were notable. Almost every VC transient after optogenetic HSN activation was accompanied by an egg-laying event, whereas only one third of VC transients during spontaneous active states were accompanied by egg-laying events. These results suggest that the high level of HSN activity after optogenetic activation induces strong coupling of VC and vulval muscle excitation.

During both spontaneous and optogenetically-induced active states, rhythmic vulval muscle twitches occur far more frequently than do VC transients, suggesting these twitches are excited by something other than the VCs. Careful observation shows that the most dramatic $Ca^{2+}$ changes during vulval muscle twitching occur at the ventral tips of the vm1 muscles (*Figure 1D*). The VCs innervate vm2 muscles but not the vm1 muscles. The vm1 muscles are instead innervated by the VB6 and VA7 neurons that also release acetylcholine to drive body wall muscle contraction for locomotion (*White et al., 1986*). Consistent with this, vm1 twitching occurs at the same time the adjacent ventral body wall muscles are contracting during locomotion (*Figure 2*). Increased electrical excitability of the vulval muscles during a spontaneous or optogenetically-induced active state may allow for vm1 excitation by VB6/VA7 with each body bend.

## VC neuron activity helps coordinate egg laying and locomotion

Our observation that VC activity always accompanies egg release (*Figures 1I* and *3A*) is consistent with a proposal that the VCs release acetylcholine that signals through vulval muscle nicotinic receptors to drive egg laying (*Waggoner et al., 2000*; *Kim et al., 2001*). While VC4 and VC5 make

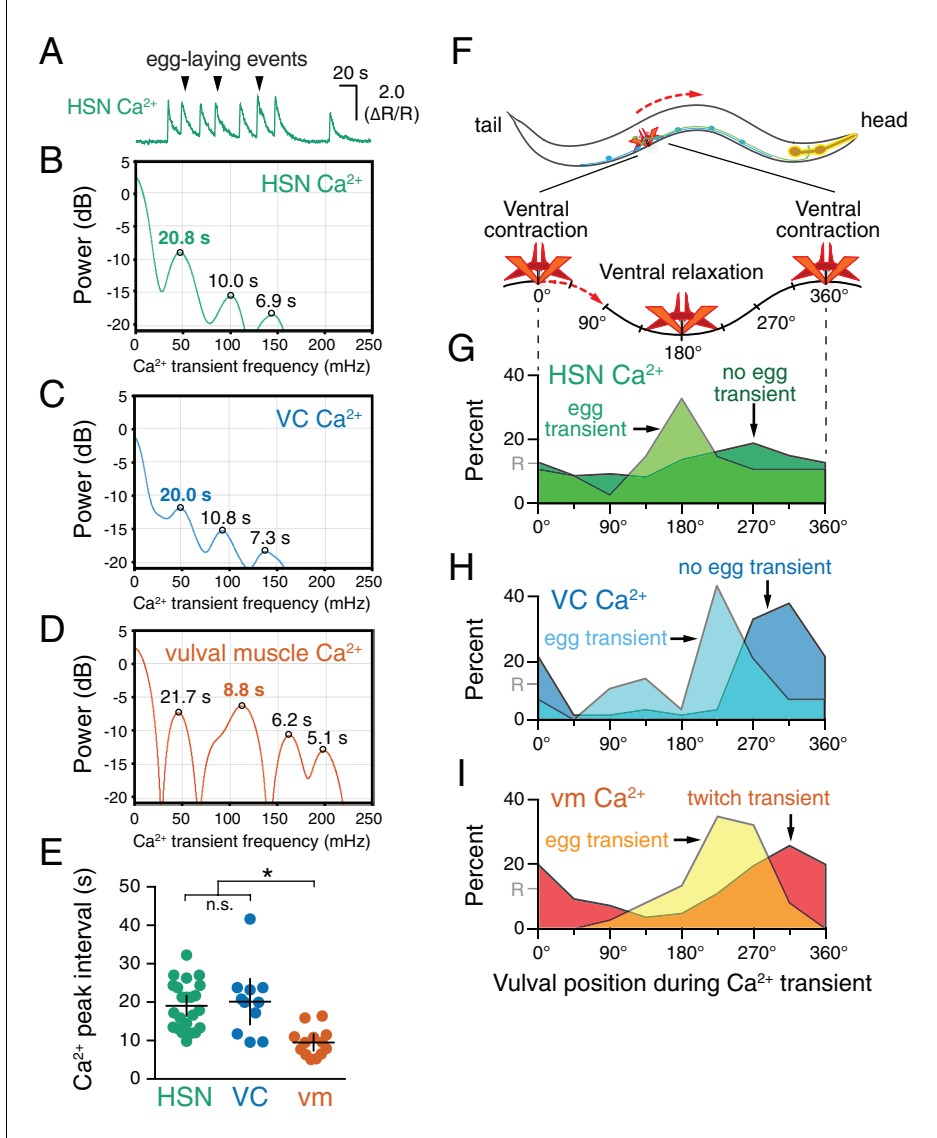

**Figure 2.** HSN, VC, and vulval muscle activity is rhythmic and phased with animal locomotion. (**A**) Active-state segments, such as the one shown here, were extracted from $Ca^{2+}$ recordings, and analyzed for rhythmicity by power spectrum analysis. Underlying rhythm frequencies and peak inter-transient intervals were thus extracted for HSN (**B**), VC (**C**), and vulval muscle (**D**) traces, and the peak of maximum rhythm for each active state is indicated in bold. (**E**) The average vulval muscle $Ca^{2+}$ peak interval (~10 s) was significantly different than the ~20 s rhythm observed in HSN (p<0.0003) and VC (p<0.0006) $Ca^{2+}$ recordings. HSN and VC rhythms were not significantly different from each other (p>0.9999; one-way ANOVA). (**F**) The position of the vulva within a sinusoidal locomotor body bend at the moment a $Ca^{2+}$ transient peaked was used to assign a body bend 'phase', in units of degrees, with 180° representing ventral relaxation and 0/360° representing ventral contraction. Plots show the percent of $Ca^{2+}$ transients observed in each of eight 45° bins for HSN (**G**), VC (**H**), and vulval muscle (**I**), with data for transients accompanied by an egg-laying event ('egg transient') plotted in different colors from data for transients not accompanied by an egg-laying event ('no egg transient'). These plots show data pooled from recordings of 8–11 animals, and *Figure 2—figure supplement 1* shows plots for each animal separately. The phasing of VC and vulval muscle transients that did not lead to egg laying is significantly different from an equal number of randomly distributed events (R, at 12.5%; p<0.0001; Kruskal-Wallis test).

The following figure supplement is available for figure 2:

**Figure supplement 1.** Relative timing of HSN, VC, and vulval muscle $Ca^{2+}$ transients during locomotion body bends.

extensive synapses onto the vm2 muscles, VC1-3 and VC6 make extensive, distributed synapses onto the ventral body wall muscles (*White et al., 1976*). How VC activity during egg laying might affect body wall muscle contraction has not been examined. We optogenetically activated the HSNs to induce active states and recorded animal speed around egg-laying events (*Figure 3C*). We found a clear and significant reduction of locomotion speed beginning ~1 s before egg release, a time when VC activity is increasing (*Figure 1I*). Locomotion speed recovered immediately after egg release (*Figure 3C*). These results are consistent with the hypothesis that acetylcholine released from the VCs slows locomotion during egg laying.

To test directly how VC activity regulates locomotion and egg laying, we used a *lin-11* promoter fragment to express Channelrhodopsin-2 specifically in the VCs (*Bany et al., 2003*). Optogenetic activation of the VCs was not sufficient to drive egg laying (*Figure 3F*), but did lead to a rapid shortening of the body (*Video 5* and *Figure 3D*) and cessation of locomotion (*Figure 3E*). The shortening phenotype seen in VC ChR2 animals after blue light exposure closely resembles that seen for optogenetic activation of the body wall muscles (*Nagel et al., 2005*), except the head and tail of VC-activated animals were still mobile and active, consistent with the fact that VC neurons innervate only the central body wall muscles (*White et al., 1976*). Our results indicate that while VC activity immediately precedes egg-laying events, it is not sufficient to drive egg release. Optogenetic VC activation is sufficient to produce a slowing of locomotion as seen during egg-laying events. In contrast, HSN activity is sufficient to induce all of the hallmarks of the active state, including rhythmic vulval muscle twitching, VC transients, and the coordinated vulval muscle contractions that produce egg-laying events.

## Animals lacking HSNs still enter the egg-laying active state

We next tested whether HSN activity was necessary for circuit activity by recording vulval muscle activity in *egl-1(dm)* mutants, which lack HSN neurons (*Trent et al., 1983*). Animals lacking HSNs have a substantial lengthening of the inactive state (*Waggoner et al., 1998*) and are strongly egg-laying defective, accumulating ~50 eggs in the uterus rather than the 12–15 seen in the wild type (*Trent et al., 1983*). We expected to find that loss of HSNs led to a dramatic reduction in vulval muscle activity. Surprisingly, we found the opposite; animals lacking HSNs had more frequent vulval muscle $Ca^{2+}$ transients during the inactive state and robust activity during infrequent active states that led to multiple eggs being laid in a short period (compare *Figure 4A and B*; quantitation in *Figure 4C*). There was a significant reduction in the amplitude of vulval muscle $Ca^{2+}$ transients during egg laying in *egl-1(dm)* mutants while the amplitude of twitch $Ca^{2+}$ transients was unaffected (*Figure 4D*). Vulval muscle activity in *egl-1(dm)* mutants lacking HSNs resembled the frequent yet uncoordinated vulval muscle activity found in animals missing the vm2 vulval muscle arms onto which the HSNs normally make synapses (*Li et al., 2013*). Such activity involves asynchronous contraction of the anterior and posterior vulval muscles, which is ineffective at allowing egg release. One quantitative measure of this asynchrony is that overall vulval muscle $Ca^{2+}$ transient peaks in *egl-1(dm)* mutants were longer in duration, likely resulting in individual muscle cells beginning their transients at different times (*Figure 4E*). Vulval muscle $Ca^{2+}$ transients in animals lacking HSNs remained phased with locomotion, including the ~45° advance in phase of vulval muscle activity that results in egg-laying events (*Figure 4—figure supplement 1*). These results suggest that the HSNs are necessary to coordinate contraction of anterior and posterior vulval muscles, and that signals from other cells besides the HSNs can also induce the egg-laying active state.

We hypothesized that accumulating unlaid eggs could stretch the uterus or body, leading to signals responsible for the increased activity of the vulval muscles independent of neurotransmitter signaling from the HSNs. To test if egg accumulation contributed to the onset of the active phase, we sterilized animals with FUDR (*Mitchell et al., 1979*), a DNA synthesis inhibitor that blocks germ line cell division and thus egg production. FUDR-treated animals had a dramatic reduction of vulval muscle activity and appeared to be stuck in the inactive state, with no egg-laying transients observed (*Figure 4A and B*). The inhibition of vulval muscle activity by FUDR occurred in both wild-type animals and HSN-deficient *egl-1(dm)* mutants (*Figure 4C and D*). The residual vulval muscle $Ca^{2+}$ twitch transients in FUDR-treated animals had phasing identical to those seen in the inactive state of untreated animals (*Figure 4—figure supplement 1*). These results suggest that germ line activity and/or the accumulation of eggs in the uterus is required to initiate the egg-laying active state.

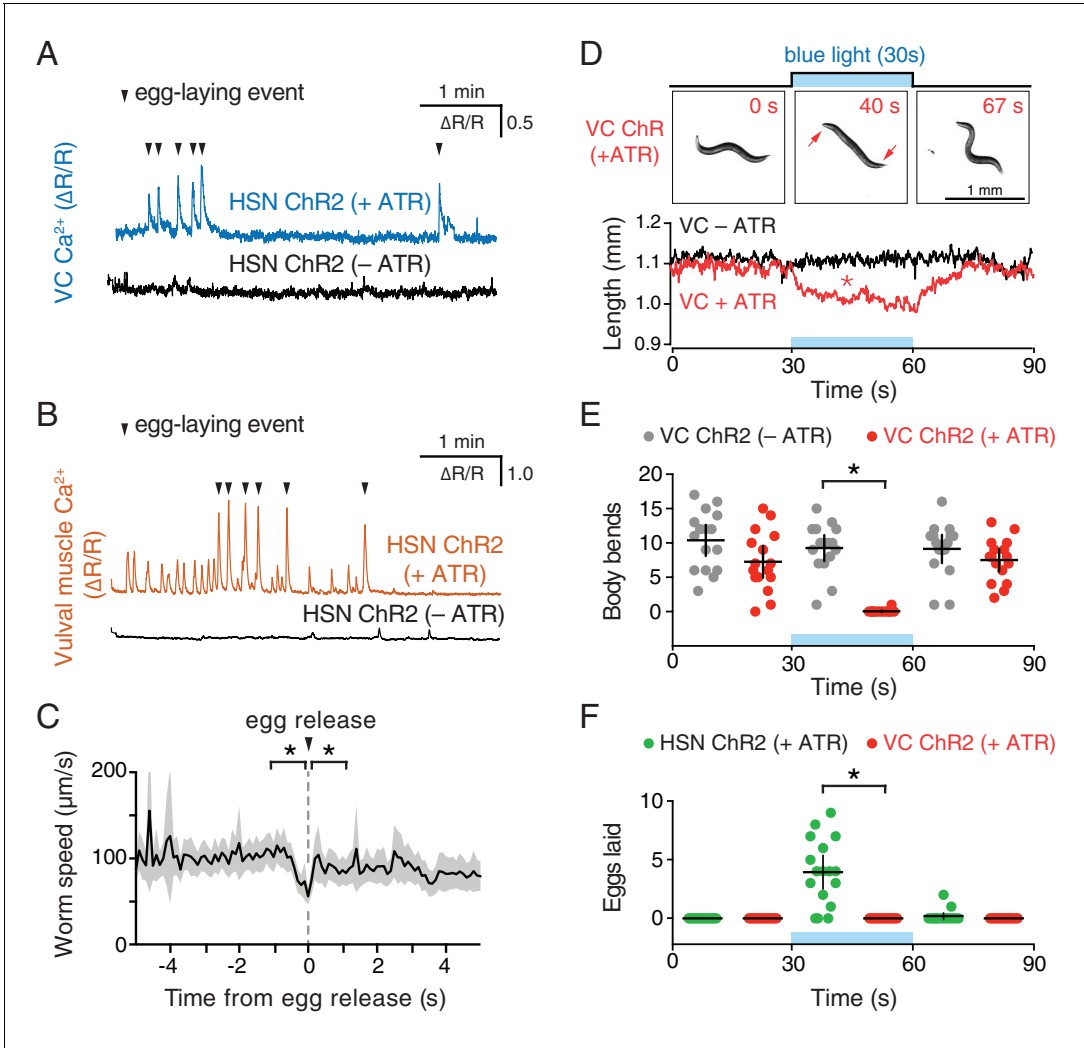

**Figure 3.** Optogenetic activation of HSN neurons initiates the egg-laying active state, and optogenetic activation of VC neurons slows locomotion. (A) Animals expressing Channelrhodopsin-2 (ChR2) in the HSNs and GCaMP5/mCherry in the VC neurons were grown in the presence (+ATR; top, blue) or absence (–ATR; bottom, black) of all-*trans* retinal. 489 nm laser light was used to simultaneously stimulate ChR2 activity and excite GCaMP5 fluorescence during the entire recording. Arrowheads indicate egg-laying events. (B) The same experiment as (A) except that GCaMP5/mCherry were expressed in the vulval muscles rather than VCs. (C) Brief reduction of animal speed during optogenetically-induced egg laying. The egg-laying active state in animals expressing ChR2 in the HSN neurons was induced with 30 s exposure to blue light. Average speed (black trace) about each egg-laying event (0 s) was calculated from the worm centroid position, and the grey area shows the 95% confidence interval. Average speed at the moment of egg laying release (0 s) is reduced compared to 1 s before or after (p<0.001). See also *Video 4*. (D) Activation of the VC neurons slows locomotion but fails to induce egg laying. Animals expressing ChR2 in the VCs were grown in the presence (+ATR; red) or absence (–ATR; black/grey) of all-*trans* retinal, and worm length (upper photos and graph) was determined. Micrographs are still images from a representative animal at indicated time points (see *Video 5*). Inset arrows indicate regions of the head and tail that remain mobile after VC activation. Scatter plots of body bends (E) and egg laying (F) from the same recordings shown in D. There was a significant change in worm length and body bends, but not egg laying, during blue light in animals grown on ATR (red; p<0.0001). Positive control animals expressing ChR2 in the HSNs (green) showed a significant increase in egg laying during blue light (p<0.0001).

## The uv1 neuroendocrine cells are mechanically deformed and activated by egg-laying events

We previously identified a role for the uv1 neuroendocrine cells in inhibiting egg-laying behavior (*Jose et al., 2007*). The four uv1 cells sit at the junction between the uterine lumen and the vulval canal (*Newman and Sternberg, 1996*), and as such are positioned to mechanically sense the accumulation of unlaid eggs in the uterus and/or passage of eggs through the vulval canal as they are

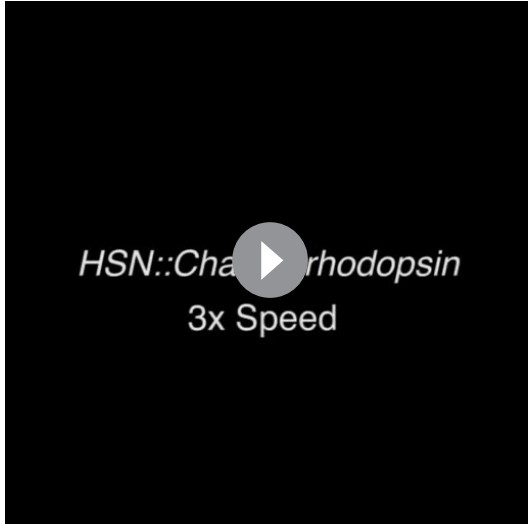

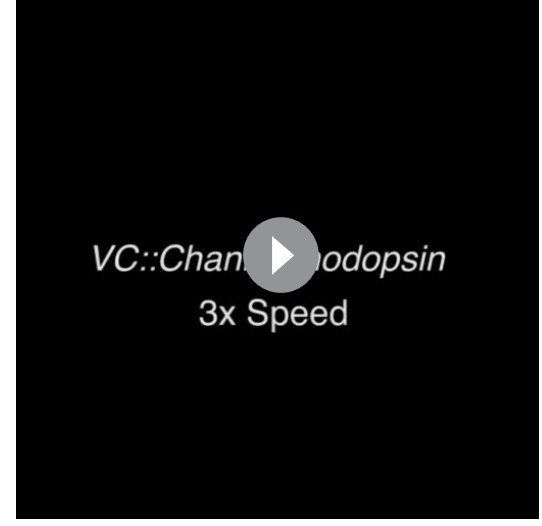

**Video 4.** Activation of the HSNs using Channelrhodopsin-2 induces the egg-laying active state. Animals were recorded for a total of 90 s with continuous blue light stimulation beginning at 30 s and ending at 60 s, during which five eggs are laid. Recording is sped up three-fold.

**Video 5.** Activation of the VCs using Channelrhodopsin-2 induces animal paralysis and shortening of body length. Animals were recorded for a total of 90 s with continuous blue light stimulation beginning at 30 s and ending at 60 s. Recording is sped up 3-fold.

laid. To record uv1 $Ca^{2+}$ activity, we co-expressed GCaMP5 and mCherry in uv1 from the *ocr-2* promoter and made recordings in behaving animals. We found that the uv1 cells were mechanically deformed as each egg passed through the vulva, and then showed strong $Ca^{2+}$ transients after egg release (*Figure 5A–B*; *Video 6*). Egg-laying events did not always trigger $Ca^{2+}$ transients in uv1, with uv1-silent events usually being the first or last within an active state (data not shown). We also observed weaker uv1 $Ca^{2+}$ transients a few minutes before or after the egg-laying active state (*Figure 5B–D*). Because these were not closely associated with egg-laying events their functional importance remains unclear. Typical active state uv1 $Ca^{2+}$ transients began immediately with or slightly before egg release, coinciding with the initial mechanical deformation of uv1 caused by the egg, with the $Ca^{2+}$ transient peak occurring ~2 s later (*Figure 5E*). Based on the position of the uv1 cells, their morphological deformations during passage of eggs, and the timing of uv1 $Ca^{2+}$ transients, our data suggest the uv1 cells are mechanically activated by eggs passing through the vulva during egg laying.

**Video 6.** Ratio recording of uv1 $Ca^{2+}$ transients during the egg-laying active state. High $Ca^{2+}$ is indicated in red while low calcium is in blue. Large panel is an expanded view showing the freely-moving animal. Inset is cropped to a small area containing the uv1s, and stabilized to remove movement. Microscope is focused to image the two uv1 cells on the right side of this animal, while the two additional uv1 cells on the left side are out of focus and not visible. Text labels indicate when egg releases occur, each of which is followed by a uv1 $Ca^{2+}$ transient.

## Tyramine signals through LGC-55 receptors to inhibit HSN activity and egg laying

The uv1 cells make the biogenic amine neurotransmitter tyramine, and *tdc-1* mutations that disrupt tyramine biosynthesis lead to increased egg-laying behaviors (*Alkema et al., 2005*). We

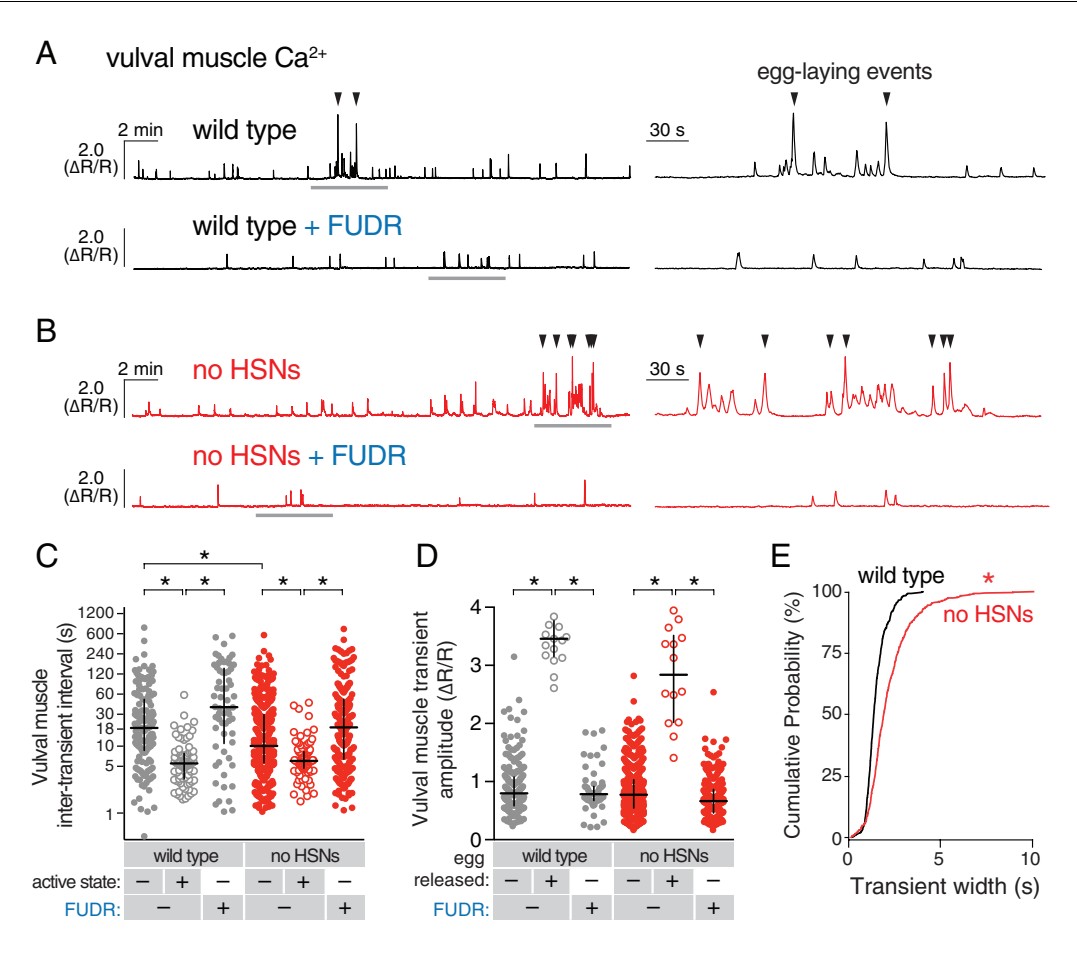

**Figure 4.** Active states require egg production but not the HSNs. (A) Vulval muscle activity in wild-type animals either untreated or after sterilization with floxuridine (FUDR). Left panel shows 30 min recordings with the grey underlined regions expanded at right. (B) Vulval muscle activity in *egl-1* (*n986dm*) mutants lacking HSNs. (C–D) Scatter plots of vulval muscle inter-transient intervals (C) and amplitudes (D). Line indicates median, error bars indicate quartiles, and asterisks indicate significant differences (p<0.01). (E) Vulval muscle transients are wider in animals lacking HSNs. Cumulative distribution plots of Ca²⁺ transient peak widths; asterisk indicates p<0.0001 (Mann-Whitney test).

The following figure supplement is available for figure 4:

**Figure supplement 1.** Persistent phasing of rhythmic vulval muscle activity in animals lacking HSNs and after sterilization.

found that exogenous tyramine reduced egg laying in wild-type animals (*Figure 6A*). To identify targets of tyramine inhibition, we tested response to exogenous tyramine in strains mutant for three known *C. elegans* tyramine receptors. Animals lacking *lgc-55*, which encodes a tyramine-gated Cl⁻ channel, showed resistance to inhibition by exogenous tyramine (*Figure 6A*), although tyramine at high levels still had a residual ability to reduce egg laying in *lgc-55* mutants (*Figure 6B*). We transformed into *C. elegans* a fosmid transgene containing the full-length *lgc-55* gene fused to GFP coding sequences, and found LGC-55 is expressed in the HSNs, uv1, and vulval muscles (*Figure 6C* and *Pirri et al., 2009*). Re-expression of LGC-55 specifically in HSN from the *tph-1* promoter restored tyramine sensitivity to *lgc-55* mutants (*Figure 6D*). We have previously shown that two Cl⁻ extruding transporters, KCC-2 and ABTS-1, are expressed in the HSNs where they promote the development of inhibitory ligand-gated Cl⁻ channel signaling (*Tanis et al., 2009*; *Bellemer et al., 2011*). These data suggest that tyramine signaling through LGC-55 would hyperpolarize the HSN and inhibit activity. To test this directly, we compared HSN activity in wild-type and *lgc-55* mutant animals. We observed a significant increase in the frequency of Ca²⁺ transients in HSNs of *lgc-55* mutant animals

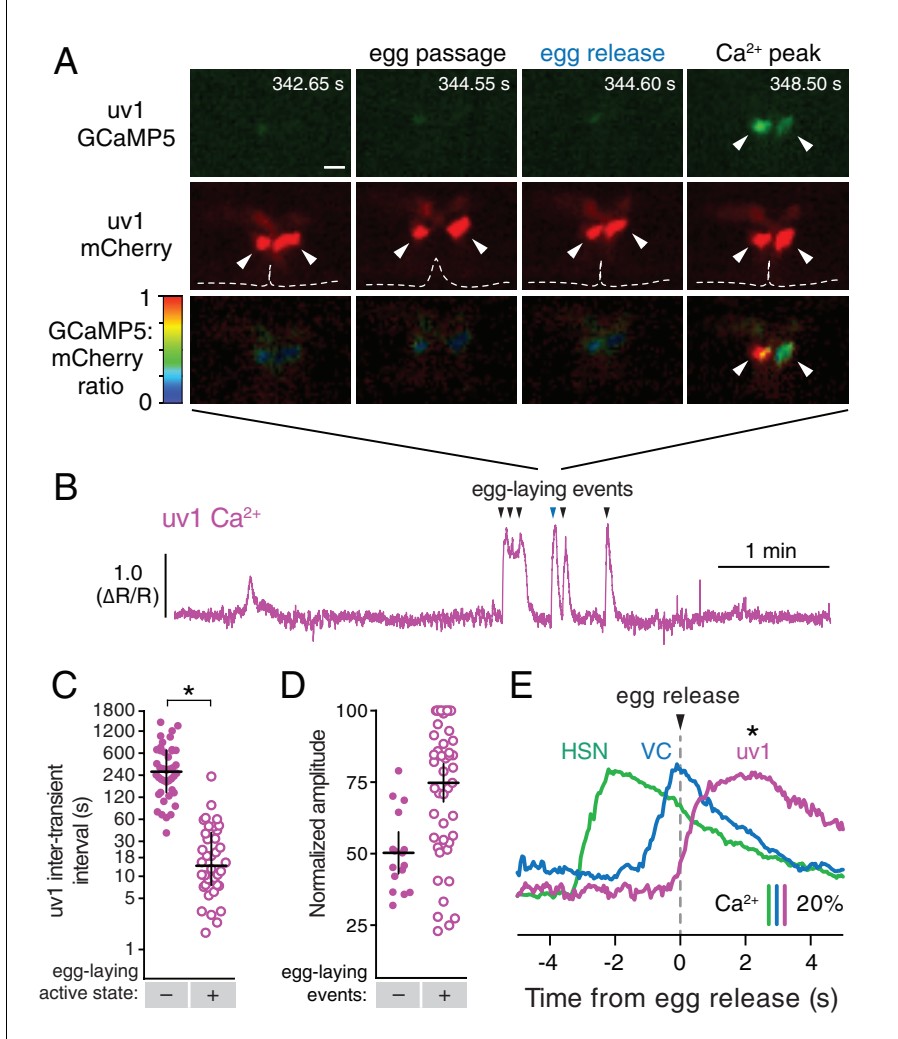

**Figure 5.** The tyraminergic uv1 neuroendocrine cells are mechanically deformed and activated by egg laying. (**A**) Fluorescence micrographs of uv1 showing GCaMP5, mCherry, and the GCaMP5/mCherry ratio before egg laying, during egg passage through the vulva, and after egg release. Times of movie frames in seconds are at top, and white scale bar is 10 μm. Arrowheads in mCherry micrographs indicate position of uv1 cells as they are deformed by egg passage, the dotted line indicates position and opening of the vulva. (**B**) GCaMP5/mCherry ratio (ΔR/R) trace of uv1 activity, including an egg-laying active state. Egg-laying events are indicated by arrowheads. (**C**) Scatter plots of uv1 inter-transient intervals during the inactive (closed circles) and active (open circles) egg-laying behavior states. Line indicates median, error bars indicate quartiles, and asterisks indicate significant differences ($p < 0.0001$). (**D**) Scatter plots and medians of normalized amplitude with (+; open circles) and without (–; closed circles) an accompanying egg release. Error bars indicate quartiles, and asterisks indicate significant differences ($p < 0.0001$, Mann-Whitney test). (**E**) uv1 $Ca^{2+}$ transients follow egg-laying events. Normalized traces of median $Ca^{2+}$ in HSN (green) and VC (blue) from *Figure 1* are compared to uv1 (purple), synchronized to the moment of egg release (0 s, arrowhead and dotted line). Bars indicate 20% change in median GCaMP5/mCherry ratio. Asterisk indicates the uv1 $Ca^{2+}$ peak is significantly later than the HSN and VC peaks ($p < 0.0001$).

(*Figure 6E and F*) in both the inactive and active states of egg-laying behavior. Mean HSN inter-transient intervals in wild-type animals were 41 ± 5 s in the inactive state and 17 ± 2 s during the active state, while intervals in *lgc-55* mutants were reduced to 22 ± 2 s in the inactive state and 13 ± 1 s during the active state. Thus, the absence of inhibitory feedback by tyramine signaling onto the HSNs leads to increased activity in both the active and inactive egg-laying behavior states.

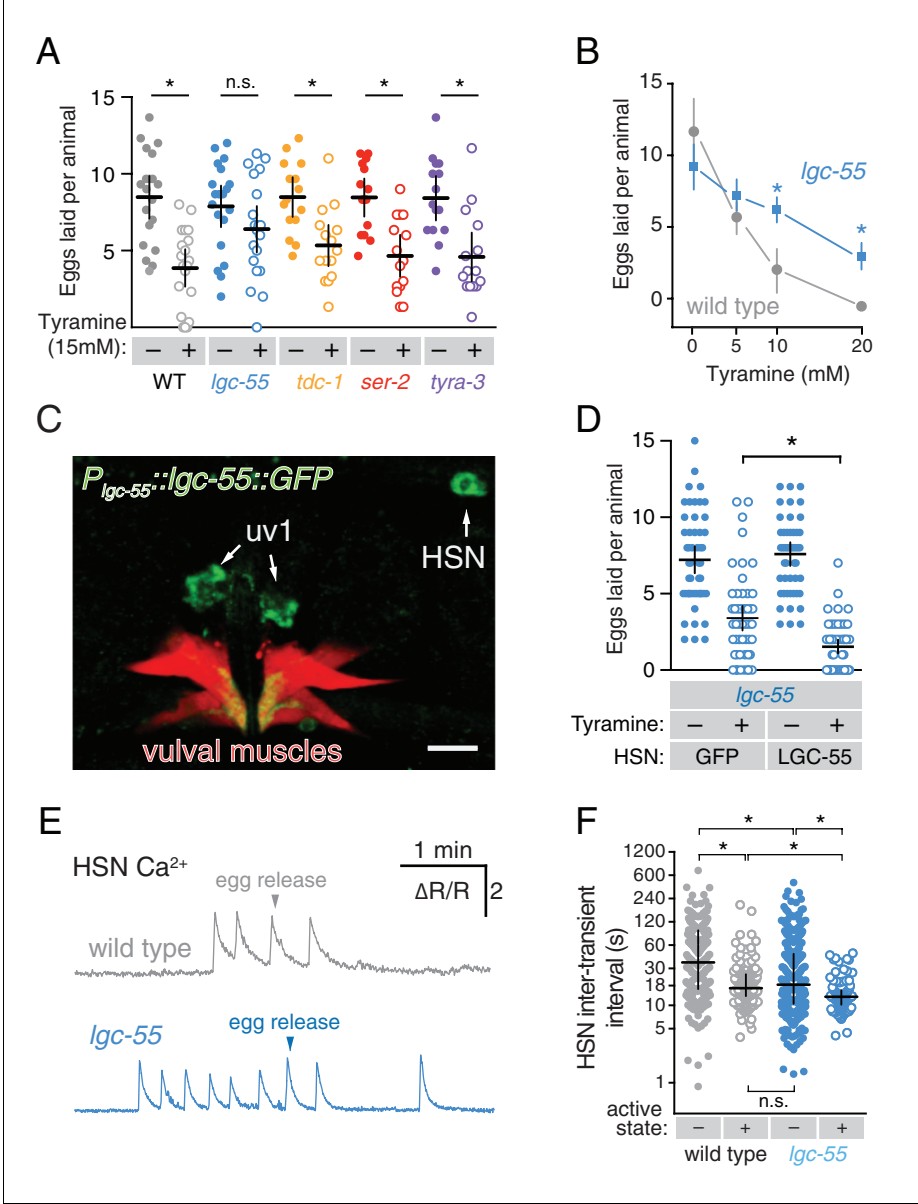

**Figure 6.** Tyramine inhibits egg laying, in part, through LGC-55 receptors on the HSNs. (A) Exogenous tyramine inhibits egg-laying behavior and requires the LGC-55 receptor. Scatter plots and means showing the average number of eggs laid after 30 min by wild type (grey), *tdc-1(n3419)* tyramine biosynthetic enzyme mutant (blue), and *lgc-55(tm2913)*, *ser-2(pk1357)*, and *tyra-3(ok325)* tyramine receptor mutant animals (orange, red, and purple, respectively) on plates without (–, closed circles) or with (+, open circles) 15 mM tyramine. Error bars indicate 95% confidence intervals, and asterisks indicate significant responses (p<0.01) while n.s. is not significant (p>0.05). (B) Dose-response curve of tyramine inhibition on egg laying in wild-type and *lgc-55* mutant animals. Asterisks indicate significant differences at 10 mM (p<0.0001) and 20 mM tyramine (p=0.0164). (C) LGC-55 is expressed in HSN, uv1, and vulval muscles. Expression of LGC-55 tagged with GFP compared to vulval muscle mCherry was visualized using confocal microscopy; scale bar is 10 μm. GFP is localized to the HSN and uv1 cell bodies (arrows) as well as the ventral tips of the vulval muscles. (D) Re-expression of LGC-55 in HSN restores tyramine inhibition of egg laying. Scatter plots of eggs laid by *lgc-55* mutants expressing either GFP or *lgc-55* in the HSN from the *tph-1* promoter. Asterisk indicates significant differences in egg laying (p=0.0029). (E) Loss of LGC-55 increases HSN activity. Representative Ca$^{2+}$ ratio traces show HSN activity in wild type (top, grey) and *lgc-55* mutant animals (bottom, blue) during the active state. (F) Scatter plots and medians of HSN inter-transient intervals in wild type and *lgc-55* mutant animals during the egg-laying inactive (–, closed circles) and active (+, open circles) states. Asterisks indicate statistically significant differences (p<0.0001).

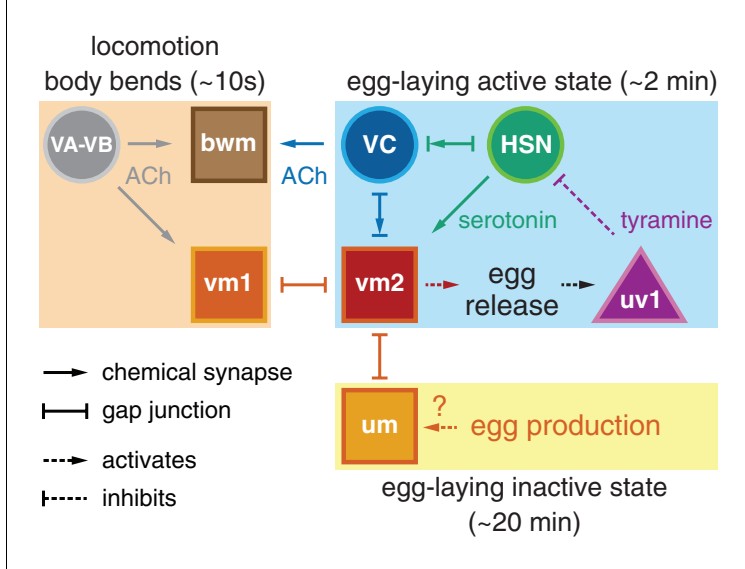

**Figure 7.** Working model of how circuit connectivity, signaling, and activity contribute to the observed rhythms that accompany the active and inactive egg-laying behavior states. VA and VB motor neuron synapses release acetylcholine (ACh) that rhythmically excites the body wall muscles (bwm) and vm1 vulval muscles during each locomotion body bend, every ~10 s. Eggs are produced from each gonad arm every 10 min, and we propose the accumulation of 2–3 eggs mechanically excites the uterine muscles (um) facilitating exit from the inactive state. Sustained bursts of HSN command motor neuron activity trigger serotonin release that drives ~2 min active states. HSN also makes synapses and gap junctions with the cholinergic VC motor neurons, and VC makes synapses and gap junctions with the vm2 muscles whose contraction allows egg release. Passage of eggs mechanically activates the uv1 cells triggering release of tyramine and neuropeptides, which inhibit HSN activity to terminate the active state. See text for further details.

## Discussion

Using a combination of optogenetic, molecular genetic, and Ca$^{2+}$ imaging approaches, we show an unexpected level of coordination of *C. elegans* egg-laying circuit activity by the rhythm of locomotion and by egg production. Using recordings of cell activity in behaving animals as they transition from the inactive to active egg-laying states, we were able to show that circuit activity is diminished during the inactive state, strongly increased and rhythmic during the active state, and this rhythm is phased with sinusoidal locomotion. Sterilization eliminated the active state rhythm, locking animals in the inactive state. Our results suggest that activity in the egg-laying circuit, like that of many other modulated motor circuits, responds to rhythmic input from central pattern generators (*Marder et al., 2015*).

A working model shown in *Figure 7* explains how different inputs into the egg-laying circuit are responsible for the inactive state and the active state rhythms observed during egg-laying behavior. Active state vulval muscle Ca$^{2+}$ transients peak rhythmically each time the vulva approaches a ventral body bend, the point at which the cholinergic VB and VA motor neurons release acetylcholine to rhythmically excite both the vm1 muscles and adjacent ventral body wall muscles during locomotion. During the inactive state, K$^+$ channels including ERG depress vulval muscle excitability below threshold, explaining why we fail to observe vulval muscle activity with each body bend during this state (*Collins and Koelle, 2013*). While we observe rhythmic activity in the HSNs in the active phase, the periodicity and phasing of HSN activity does not correlate with that of locomotion. Consistent with this, previous work found changes in osmolarity can drive rhythmic HSN activity even in immobilized animals (*Zhang et al., 2008*). Thus HSN rhythmic activity is independent of locomotion, although we found HSN Ca$^{2+}$ transients that happen to fall at a specific phase of locomotor body bends are more likely to be followed by egg release. We propose that release of serotonin from the HSNs, along with an unknown signal resulting from mechanical stretch of the uterus by accumulating eggs,

together raise vulval muscle electrical excitability to result in the active state during which the vulval muscles undergo rhythmic twitching or egg laying contractions in response to the same cholinergic VA and VB neurons that drive locomotor body bends. This model accounts for the phasing of vm1 transients with body bends, although egg-laying events require additional activity in the vm2 muscles that results in the coordinated contraction of all vm muscles that results in egg release. We found that vm transients that lead to egg release are on average slightly phase advanced relative to those that do not. This phase advance could from the excitatory input onto the vm2 muscles occurring earlier in the body bend, and candidates for this input are the HSNs and VCs, both of which synapse onto the vm2 muscles and have phased activity during egg-laying events. The functional purpose of this phase advance may be mechanical. Passage of each $25 \times 40$ µm egg requires contraction of vm1 and vm2 muscles on both the anterior and posterior sides of the vulva (*Li et al., 2013*). There may not be sufficient room for vulva to fully open and release egg unless the adjacent body wall muscles are in a contracted state.

The HSNs are command neurons that integrate diverse sensory inputs and initiate the ~2 min active state of the egg-laying circuit. Optogenetic activation of HSNs recapitulates features of the active state including phased VC and vulval muscle activity and multiple egg-laying events. However, stimulation of the HSNs fails to drive tetanic contraction of the vulval muscles as would be predicted for a motor neuron releasing a fast-acting neurotransmitter. Instead, HSN makes and releases the neuromodulator serotonin which signals through G protein coupled serotonin receptors to activate the vulval muscles (*Desai et al., 1988*; *Shyn et al., 2003*; *Carnell et al., 2005*; *Hobson et al., 2006*; *Hapiak et al., 2009*). The HSNs also release serotonin onto the AVF interneurons in the nerve ring to promote a state of locomotion arousal that accompanies the active state (*Hardaker et al., 2001*). Animals defective in serotonin signaling have reduced egg laying (*Tanis et al., 2008*), but this defect is not as severe as that of *egl-1(dm)* animals missing HSNs (*Trent et al., 1983*), suggesting HSN releases additional neurotransmitters. We found that optogenetic activation of HSNs leads to rhythmic activity of VC neurons, which are not known to express serotonin receptors, so an unknown signal derived from HSN could directly activate VCs, as depicted in the *Figure 7* model.

How is HSN activity controlled? The HSNs receive little synaptic input near the vulval except from the PLM mechanosensory neurons (*White et al., 1986*). The PLMs inhibit HSN activity and egg laying in response to gentle touch (*Zhang et al., 2008*), but recent work indicates the PLM neurons are also deformed during normal patterns of locomotion that could contribute to rhythmic activity seen in HSN during egg laying (*Topalidou et al., 2012*). Neuroendocrine signaling also regulates HSN activity. *flp-10* and *flp-17* neuropeptides are released from the $CO_2$-responsive BAG neurons in the head, and these neuropeptides signal through EGL-6 receptor and $G\alpha_o$ to activate IRK channels that inhibit HSN activity and egg laying (*Ringstad and Horvitz, 2008*; *Hallem et al., 2011*; *Emtage et al., 2012*). Because loss of EGL-6 or IRK channels do not increase egg laying as much as cell-specific inhibition of $G\alpha_o$, there must be additional receptors and effectors of $G\alpha_o$ that inhibit HSN activity. $G\alpha_q$ signaling opposes $G\alpha_o$ to increase serotonin release from the HSNs (*Tanis et al., 2008*), but the ligands and receptors that activate $G\alpha_q$ have not yet been identified. Because we find that HSN $Ca^{2+}$ transients vary more in frequency than amplitude, we propose that these diverse sensory signals increase or reduce the probability that HSN reaches a threshold potential that can initiate, sustain, or terminate HSN activity and thus the active behavior state of the egg-laying circuit.

We have also identified an HSN-independent mechanism that induces the active state. Mutants lacking HSNs are strongly egg-laying defective, but eventually enter active states with properly phased vulval muscle twitching and egg-laying transients. The active state was eliminated in animals sterilized by FUDR, reducing vulval muscle activity and phasing to that seen in the inactive state. Recent work has identified the gonad as an important site of regulation for adult-specific behaviors in *C. elegans* (*Fujiwara et al., 2016*). We propose that the accumulation of unlaid eggs in the uterus, perhaps leading to mechanical stretch, provides an essential, permissive signal for entry into the active state. Several steps during embryo production and maturation could regulate entry into the active state to result in the ~20 min period of the egg-laying inactive state (*Waggoner et al., 1998*). Animals grown on food generate one new oocyte from each of two gonad arms every ~10 min, balancing out the 3–5 eggs laid in the ~2 min active state that occurs every ~20 min, to result in a ~constant number of unlaid eggs in the uterus (*McCarter et al., 1999*). New embryos are moved by gonadal sheath contractions into the spermatheca for fertilization and deposition into the uterus.

Excitable cells drive each of these steps, providing potential sources of regulation by cell signaling and sensory feedback (*Kariya et al., 2004*). Eight uterine muscles line the uterine lumen and form gap junctions with the vm2 muscles (*White et al., 1986*), and the addition of unlaid eggs into the uterus could induce a mechanical stretch that cross-excites the vm2 muscles. Future work will be required to understand how the accumulation of unlaid eggs promotes the active state.

Our results strongly imply that acetylcholine released from the VC motor neurons slows locomotion during vulval muscle contraction. We found that animal locomotion slows during egg release, and that optogenetic activation of the VC neurons halts movement and results in body wall muscle contraction and animal shortening. This function of the VCs is likely via acetylcholine signaling through nicotinic receptors expressed on ventral body wall muscles and neurons that regulate locomotion (*White et al., 1986*; *Nagel et al., 2005*). The effect of VC activity to slow locomotion and contract body wall muscles may help compress the body and uterus to encourage expulsion of eggs, analogous to the role of body wall muscle contraction in facilitating defecation (*Liu and Thomas, 1994*).

Our results provide more complex evidence as to whether VC acetylcholine also directly triggers egg release. We found that VC activity always accompanies egg-laying contractions of the vulval muscles. The vulval muscles also express nicotinic acetylcholine receptors, and receptor agonists including levamisole and nicotine stimulate egg laying (*Weinshenker et al., 1995*; *Waggoner et al., 2000*). However, we found little correlation between the strength of the VC activity and whether an egg is laid (*Figure 1H*), and that optogenetic activation of the VC neurons fails to drive egg laying. Further, VC-defective and acetylcholine-deficient mutants have *increased* egg-laying behavior (*Bany et al., 2003*; *Ringstad and Horvitz, 2008*). These data together are consistent with a model in which acetylcholine released from the VA/VB motor neurons signal through nicotinic receptors on the vm1 muscles to stimulate rhythmic vulval muscle twitching, which combined with acetylcholine released from the VC neurons to similarly excite the vm2 muscles results in egg laying. High levels of VC activity, such as occurs during optogenetic activation, may drive release of additional signals, such as neuropeptides, that inhibit egg-laying circuit activity (*Bany et al., 2003*; Banerjee and Francis, personal communication), or by freezing locomotion may feedback to halt the VA/VB activity required to drive vm1 contraction.

How is VC activity controlled? The VC neurons are unusual in that they are largely presynaptic, only receiving input from each other and the HSNs. As noted above, optogenetic activation of HSNs results in VC activity, suggesting a currently unknown signal from HSN directly activates VCs. A second possible input onto VCs is suggested by the fact that the VC neurons extend processes devoid of synapses dorsally along the vulval hypodermis that could be mechanosensory (*White et al., 1986*). Thus, rhythmic VC activity could be stimulated mechanically by body bends or by a humoral signal released from the locomotor circuit (*Hu et al., 2011*). Further, VC activity could be mechanically stimulated by vulval muscle contraction (*Zhang et al., 2010*), particularly as we observed mechanical deformation of VC presynaptic termini during strong twitching and egg-laying contractions. In this way, the VCs could act like baroreceptors–mechanically activated by vulval muscle contraction to drive contraction of the body wall muscles that results in animal slowing until the feedback of egg release. In this model, the coordination of locomotion and egg laying by tying both behaviors to the same central pattern generator allows productive egg release.

The uv1 neuroendocrine cells inform the circuit that eggs have been successfully laid. Passage of eggs through the vulva mechanically deforms the uv1 cells and triggers a strong $Ca^{2+}$ transient. Our results suggest this activates uv1 to release tyramine that inhibits egg-laying, at least in part via the LGC-55 receptor on the HSN neurons. Active states last ~2 min, and inhibitory tyramine release from uv1 could provide a feedback signal that terminates activity in the circuit. The four uv1 cells have a unique structural position in the vulva that may explain how they become activated during egg laying (*Newman et al., 1996*). uv1 cells extend processes that contact and make adherens junctions with the ventral vulF vulval cells. uv1 cells also makes dorsal attachments to utse, a large, H-shaped multinucleate uterine seam cell that attaches the uterus to the lateral epidermis and that functions as a hymen broken by the first egg-laying event. The acute mechanical strain on vulF, uv1, and/or utse during egg passage could mechanically activate ion channels. Previous work has shown uv1 expresses several TRPV channels that may be mechanosensory and contribute to this response (*Jose et al., 2007*). A dominant-negative OCR-2 mutant expressed in uv1 leads to hyperactive egg-laying behavior phenotype. We still observe $Ca^{2+}$ transients in uv1 after egg-laying events in this

*ocr-2(vs29)* mutant (data not shown), suggesting other channels contribute to the uv1 activity we see after egg laying. Although our results indicate that uv1 releases tyramine to inhibit the HSNs via LGC-55, this signaling event does not fully explain the ability of uv1 cells to inhibit the egg-laying circuit. We found that inhibition of egg laying by exogenous tyramine is reduced but not eliminated in *lgc-55* null mutants, suggesting other tyramine receptors help terminate the active state. The uv1 cells also make *nlp-7* and *flp-11*, two neuropeptides that inhibit HSN activity and egg-laying behavior (Banerjee and Francis, personal communication), and the identification of the receptors and signaling pathways through which they inhibit serotonin release will help to more fully understand how activity of the egg-laying circuit is terminated.

Every neural circuit may require specific signals to initiate activity when its function is required, signals to coordinate the dynamic activity of its various cells, and mechanisms to determine when the circuit has successfully completed its task and signal to turn off activity. Many circuits also must respond to specific signals from central pattern generators that coordinate the activity of multiple circuits. A great challenge is to reduce these abstract concepts to concrete, specific mechanisms in the case of a real circuit. For the *C. elegans* egg-laying circuit, signals in each of the categories outlined above are now identified, albeit in varying levels of detail and completeness. Continuing analysis of the *C. elegans* egg-laying circuit using recordings of Ca$^{2+}$ activity in behaving animals, combined with genetic, optogenetic, and pharmacological manipulation, provides a path towards detailed understanding the mechanisms that control activity of this model circuit.

**Table 1.** Strains used in this study.

| Strain | Feature | Genotype | Figures |
|--------|---------|----------|---------|
| LX1832 | Strain for transgene production | *lite-1(ce314) lin-15(n765ts)* X | 1–6 |
| LX1836 | HSN Channelrhodopsin | *wzIs30* IV; *lite-1(ce314) lin-15(n765ts)* X | 3 |
| LX1918 | vulval muscle GCaMP5, mCherry | *lite-1(ce314) vsIs164 lin-15(n765ts)* X | 1, 2 |
| LX1932 | HSN Channelrhodopsin, vulval muscle GCaMP5, mCherry | *wzIs30* IV; *lite-1(ce314) vsIs164 lin-15(n765ts)* X | 3 |
| LX1938 | No HSNs, vulval muscle GCaMP5, mCherry | *egl-1(n986dm)* V; *lite-1(ce314) vsIs164 lin-15(n765ts)* X | 4 |
| LX1986 | uv1 GCaMP5, mCherry | *vsIs177*; *lite-1(ce314) lin-15(n765ts)* X | 5 |
| LX1960 | VC GCaMP5, mCherry | *vsIs172*; *lite-1(ce314) lin-15(n765ts)* X | 1, 2 |
| LX1970 | HSN Channelrhodopsin, VC GCaMP5, mCherry | *wzIs30* IV; *vsIs172*; *lite-1(ce314) lin-15(n765ts)* X | 3 |
| LX2004 | HSN GCaMP5, mCherry | *lite-1(ce314), vsIs183 lin-15(n765ts)* X | 1, 2, 6 |
| LX2038 | *lgc-55* null mutant HSN GCaMP5, mCherry | *lgc-55(tm2913)* V *lite-1(ce314) vsIs183 lin-15(n765ts)* X | 6 |
| MIA3 | VC Channelrhodopsin | *keyIs3*; *lite-1(ce314) lin-15(n765ts)* X | 3 |
| MT13113 | *tdc-1* null mutant; no tyramine biosynthesis | *tdc-1(n3419)* II | 6 |
| N2 | Bristol strain | wild type | 6 |
| OH313 | *ser-2* null mutant | *ser-2(pk1357)* X | 6 |
| VC125 | *tyra-3* null mutant | *tyra-3(ok325)* X | 6 |
| QW89 | *lgc-55* null mutant | *lgc-55(tm2913)* V | 6 |
| LX2096 | vulval muscle mCherry | *vsIs191*; *lin-15(n765ts)* X | |
| LX2081 | pan-neuronal TagRFP | *unc-119(ed3)* I; *otIs356* | |
| LX2137 | vulval muscle mCherry, *lgc-55::gfp* | *vsIs191*; *vsEx791* | 6 |
| LX1330 | *lgc-55* mutant expressing GFP in HSN from *tph-1* promoter | *lgc-55(tm2913)* V; *lin-15(n765ts)* X *vsEx557* | 6 |
| LX1329 | *lgc-55* mutant expressing LGC-55 in HSN from *tph-1* promoter | *lgc-55(thm2913)* V; *lin-15(n765ts)* X *vsEx558* | 6 |

## Materials and methods

### Nematode culture, strains, and maintenance

*Caenorhabditis elegans* strains were maintained as hermaphrodites at 20°C on Nematode Growth Medium (NGM) agar plates with *E. coli* OP50 as a source of food as described (*Brenner, 1974*). All strains are derived from the Bristol N2 wild-type strain, and all assays were performed using age-matched adult hermaphrodites ~24 hr past the late L4 stage using recommended sample sizes (*Chase and Koelle, 2004*). Animals were treated with 0.1 mg/ml floxuridine (FUDR) at a late L4 stage when all non-germline cell divisions are complete. A list of strains, mutants, and transgenes used in this study can be found in *Table 1*.

### Molecular biology and transgenes

Vulval muscles

GCaMP5G (*Akerboom et al., 2012*) and mCherry were expressed in the vulval muscles from the *unc-103e* promoter as previously described (*Collins and Koelle, 2013*). Briefly, pKMC274 (GCaMP5, 20 ng/μl) and pKMC257 (mCherry, 2 ng/μl) were co-injected along with the pL15EK *lin-15* rescue plasmid (50 ng/μl) into LX1832 *lite-1(ce314), lin-15(n765ts)* X animals. The GCaMP5/mCherry extra-chromosomal transgene produced was chromosomally integrated using UV/TMP mutagenesis, creating the integrated transgene *vsls164*, and the resulting transgenic animals were backcrossed to the LX1832 parental line six times.

HSN

GCaMP5G and mCherry were expressed in the HSN motor neurons using the *nlp-3* promoter. mCherry and GCaMP5G coding sequences were ligated into pJB9 bearing the *nlp-3* promoter and *nlp-3* 3' untranslated region, and the resulting plasmids (GCaMP5, pKMC300, 80 ng/μl; mCherry, pKMC299, 20 ng/μl) were co-injected with pL15EK (50 ng/μl) into LX1832 worms. The extrachromo-somal transgene produced was integrated with UV/TMP, creating the integrated transgene *vsls183*, and the resulting transgenic animals were backcrossed to the LX1832 parental line six times. The *wzls30* transgene expressing Channelrhodopsin-2 from the *egl-6* promoter has been described (*Emtage et al., 2012*).

VC

GCaMP5G and mCherry were expressed in the VC motor neurons using pKMC145, a pDM4-based plasmid bearing a *lin-11* enhancer region fused to the *pes-10* basal promoter with a modified multiple cloning site (*Bany et al., 2003*; *Tanis et al., 2008*). Derivatives expressing GCaMP5 (pKMC273, 80 ng/μl) and mCherry (pKMC275, 30 ng/μl) were co-injected with pL15EK (50 ng/μl) into LX1832 worms. The extrachromosomal transgene produced was integrated with UV/TMP, creating the trans-gene *vsls172*, and transgenic animals were backcrossed to the LX1832 parental line six times. To activate the VCs optogenetically, coding sequences for ChR2-YFP bearing the H134R, T159C muta-tions (*Erbguth et al., 2012*) were amplified by PCR, ligated into pKMC145 to generate pKMC283, injected at 80 ng/μl into LX1832 along with pL15EK at 50 ng/μl. The extrachromosomal transgene produced was integrated with UV/TMP creating the transgene *keyls3*, and the resulting animals were backcrossed to LX1832 parental line six times.

uv1

GCaMP5G and mCherry were expressed in uv1 using the *ocr-2* promoter. GCaMP5 and mCherry coding sequences were ligated into pJT79 between the *ocr-2* promoter and 3' untranslated region (*Jose et al., 2007*) to generate plasmids pKMC281 and pKMC284, respectively. Long-range PCR products amplified from these plasmids were injected into worms at 80 ng/μl each along with pL15EK (50 ng/μl) as previously described (*Jose et al., 2007*). The extrachromosomal transgene pro-duced was integrated with UV/TMP creating the transgene *vsls177*, and the resulting animals were backcrossed to the LX1832 parental line six times.

## lgc-55

A ~40 kb genomic fosmid clone (WRM063D_A12) recombineered to express LGC-55 tagged with GFP at the C-terminus (*Sarov et al., 2012*) was injected at 100 ng/µl into LX2081 *unc-119(ed3)* mutant animals expressing TagRFP pan-neuronally to generate the strain LX2126. These animals were then crossed with LX2096 animals to exchange pan-neural mCherry transgene for one that expresses in the vulval muscles, generating LX2126. Fluorescence was visualized on immobilized animals with a 63x water objective on a Zeiss LSM 710 Duo confocal microscope. For cell-specific rescue, LGC-55 was expressed in the HSN neurons of *lgc-55* mutant animals using the *tph-1* promoter. Briefly, a *lgc-55* cDNA (*Pirri et al., 2009*) was inserted into pJM66A plasmid (*Moresco and Koelle, 2004*) to generate pJT105. pJT105, or a GFP-expressing control plasmid (pJM60A), was injected at 100 ng/µl along with pL15EK rescue plasmid (50 ng/µl) into *lgc-55(tm2913); lin-15(n765ts)* double mutant animals. Egg-laying behavior was analyzed in six independent transgenic lines expressing LGC-55 or GFP, and one strain of each were kept for long-term storage.

## Ratiometric Ca$^{2+}$ imaging

Animals were mounted between glass coverslips and chunked sections of NGM plates for imaging as described (*Collins and Koelle, 2013*; *Li et al., 2013*). The stage and focus were adjusted manually to keep the egg-laying system in view during recording periods. Long-term recordings were cropped to 10–30 min with the first egg-laying event at 5 min. Three different imaging systems were used to collect recordings, and each gave quantitatively similar results. First, single, two-channel confocal slices (18 µm thick) of GCaMP5 and mCherry fluorescence were collected through a 20x Plan Apochromat objective (0.8NA) using a Zeiss Axio Observer microscope and recorded at 20 fps at 256 × 256 pixel resolution, 16-bit depth using a 710 Duo LIVE scanner as described previously (*Collins and Koelle, 2013*; *Li et al., 2013*). Second, widefield GCaMP5 and mCherry fluorescence was imaged through a 20x Plan Apochromat objective (0.8NA) using a Zeiss Axio Observer microscope, and the GCaMP5 and mCherry fluorescence was separated using a W-View Gemini image splitter at 20 fps at 256 × 256 pixel resolution, 16-bit depth onto a single, 4 × 4 binned Hamamatsu ORCA Flash 4.0 V2 sCMOS sensor after excitation with 470 nm and 590 nm LEDs (Zeiss Colibri.2). Third, single, two-channel confocal slices (20 µm thick) of GCaMP5 and mCherry fluorescence were collected using an inverted Leica TCS SP5 confocal microscope with the resonant scanner running at 8000 Hz and collected at 20 fps at 256 × 256 pixel resolution, 12-bit depth after 2x digital zoom.

Quantitative ratiometric analysis was performed after importing image sequences into Volocity (Perkin Elmer). GCaMP5/mCherry ratios were calculated, and pixels with mCherry fluorescence intensities ~2 standard deviations above background were selected as objects for mean ratio measurement. While the Volocity algorithm set with these parameters typically identified the vulval muscles and uv1 cells as a single object, the complex morphology of the HSN and VC neurons often led to the cells being found as two or more objects within each video frame. These were rejoined using a custom script in MATLAB (Mathworks). The ratio of the joined object was scaled in proportion to the area of each component object to generate an area-adjusted GCaMP5/mCherry ratio. The ratio data were then smoothed using a 150 msec (three timepoint) rolling average, and the lowest 10% of the GCaMP5/mCherry ratio values over the entire recording period were averaged to establish a ΔR/R baseline. Ca$^{2+}$ transients and their features (peak amplitude and width at half-maximum) were identified from ratio traces using the *findpeaks* algorithm in MATLAB, and peaks were confirmed by visual inspection of GCaMP5/mCherry recordings and ratio traces. Egg-laying events were observed by allowing a small amount of bright field light to transmit into the mCherry channel whose gain was increased for egg visualization. The egg-laying active state was defined in recordings to include those transients that occurred one minute before and one minute after each egg-laying event. Those transients not found within an active state were considered to be in the inactive state. Inter-transient intervals were calculated by measuring the elapsed time between successive Ca$^{2+}$ transient peaks during the inactive and active state. Because some inactive states were devoid of detectable activity (e.g. *Figure 1F*, VC Ca$^{2+}$), inter-transient intervals during such periods were operationally defined as the time before the first or after the last transient in the recording. Because the recordings were cropped to focus on the egg-laying active state, the inactive state inter-transient intervals reported here should be considered a lower estimate. To facilitate comparisons of ΔR/R between different reporters and recording conditions (*Figures 1H* and *5D*), the maximum peak

amplitude was normalized to 1. Spectral analysis of active phase $Ca^{2+}$ traces was performed by Fast Fourier Transform method of estimation using the Signal Analyzer app in the MATLAB Signal Processing toolbox (R2016b). Quantitatively similar results were obtained using the Lomb-Scargle periodogram function and using auto-correlation analysis (data not shown). The largest amplitude frequency peak between 0 and 250 mHz from each active state periodogram was compared using one-way ANOVA.

## Locomotion phasing analysis

We recorded the position of the vulva during each sinusoidal body bend of locomotion relative to observed HSN, VC, and vulval muscle $Ca^{2+}$ transients. We recorded the timing of each ventral contraction and/or relaxation flanking each $Ca^{2+}$ transient peak. To approximate the phasing of that $Ca^{2+}$ relative to locomotion, we operationally defined 0° and 360° as when the vulva was positioned when the adjacent body wall muscles were at their most ventrally contracted state, and we defined 180° when the vulva was positioned when the adjacent body wall muscles were at their most relaxed state. The relative timing of each $Ca^{2+}$ transient peak or behavior event was calculated as fraction of time elapsed between the previous and subsequent contraction or relaxation, regardless of whether the animal was engaged in forward or reverse locomotion, as described previously (*Collins and Koelle, 2013*). For example, if a $Ca^{2+}$ transient peaked exactly 2 s after the vulva passed through its most ventrally relaxed state (180°) and 2 s prior to the ventral contracted state (360°), the $Ca^{2+}$ transient phasing would be halfway between 180° and 360°, or 270°. If a $Ca^{2+}$ transient peaked halfway between ventral contraction (0°) and ventral relaxation (180°), the $Ca^{2+}$ transient phasing of that peak would be 90°. This method of estimation allows for phasing information to be extracted and compared independent of differences in individual animal speed and locomotion direction.

## Optogenetics

Channelrhodopsin-2 (ChR2) expressing strains were grown in the presence or absence of the ChR2 cofactor all-*trans* retinal (ATR). ATR was prepared at 100 mM in 100% ethanol and stored at −20℃. To prepare NGM plates for behavior analysis, ATR was diluted to 0.4 mM with warmed cultures of OP50 bacteria in B Broth, and 200 µl of culture was seeded onto each 60 mm NGM plate. The plates were allowed to grow for 24 hr at 25–37℃, after which late L4 worms were staged onto prepared plates for behavioral assays 24 hr later. During $Ca^{2+}$ imaging experiments, ChR2 was activated with the same light used for GCaMP5 excitation. For timed behavior recording experiments on plates, a OTPG_4 TTL Pulse Generator (Doric Optics) was used to trigger image capture (Grasshopper 3, 4.1 Megapixel, USB3 CMOS camera, Point Grey Research) and shutter opening on a EL6000 metal halide light source generating 8–16 mW/cm$^2$ (depending on the magnification used) of ~470 ± 20 nm blue light via a EGFP filter set mounted on a Leica M165FC stereomicroscope. Worm size, speed, and body bends were determined from image sequences using Volocity.

## Tyramine behavior assays

Three adult animals were placed onto each of three 35 mm agar plates prepared with tyramine-HCl (Sigma-Aldrich, St. Louis, MO) and acetic acid seeded with OP50 bacteria as described (*Pirri et al., 2009*). The average number of eggs laid was counted after 30 min from at least 11 trials for each genotype and tyramine concentration.

## Statistical analyses

Data were analyzed using Prism 6 (GraphPad). Average peak transient rhythm, worm speed, worm length, body bends, or eggs laid were compared using one-way ANOVA with Bonferroni correction for multiple comparisons, and error bars indicate 95% confidence intervals for the mean. $Ca^{2+}$ transient peak amplitudes, widths, and inter-transient intervals were pooled from multiple animals (n $\geq$ 4 animals per genotype/condition per experiment), compared using the Mann-Whitney test (for pairwise comparisons) or the Kruskal-Wallis test with Dunn's correction (for multiple comparisons), and error bars indicate quartile intervals for the indicated median. p<0.05 interactions were deemed significant and are labeled with an asterisk. *p* values for specific interactions are indicated in the Figure legends. Sample sizes for behavioral assays followed previous studies (*Waggoner et al., 1998*; *Chase and Koelle, 2004*; *Collins and Koelle, 2013*). The number of animals analyzed in each

experiment is indicated below, with each $Ca^{2+}$ transient or egg-laying event serving as a biological replicate.

### Figures 1, 2, and Figure 2–figure supplement 1

HSN, 235 inactive and 157 active state intervals between 319 no-egg and 49 egg-laying transients from 11 animals; VC, 22 inactive and 83 active state intervals between 59 no-egg and 32 egg-laying transients from eight animals; vulval muscles, 73 inactive and 136 active state intervals between 175 twitch (no-egg) and 37 egg-laying transients from seven animals.

### Figure 3

A, ±ATR, seven animals; B, -ATR, nine animals; +ATR, 10 animals; C, 52 egg-laying events from 17 HSN-ChR2 +ATR animals; D-F (all genotypes), -ATR, 18 animals; +ATR, 17 animals.

### Figure 4

A-E, wild type -FUDR, 150 inactive and 87 active state intervals between 221 twitch (no-egg) and 17 egg-laying transients from four animals; wild-type +FUDR, 62 inactive state intervals between 59 twitch (no-egg) transients from four animals; *egl-1(n986dm)* -FUDR, 450 inactive and 97 active state intervals between 531 twitch (no-egg) and 14 egg-laying transients from seven animals; *egl-1 (n986dm)* +FUDR, 228 inactive state intervals between 222 twitch (no-egg) transients from six animals.

### Figure 4–figure supplement 1

A-C, 239 twitch (no-egg) transients and 22 egg-laying transients from five untreated wild-type animals (A) and 91 twitch (no-egg) transients from 5 FUDR-treated wild-type animals (B); D-F, 550 twitch (no-egg) transients and 18 egg-laying transients from seven untreated *egl-1(n986dm)* animals (D) and 232 twitch (no-egg) transients from 8 FUDR-treated *egl-1(n986dm)* animals (E).

### Figure 5

uv1, 24 inactive and 42 active state intervals for 16 no-egg and 51 egg-laying transients from 13 animals.

### Figure 6

A, wild-type, 60 animals analyzed per tyramine concentration, three animals on each of 20 plates containing either water or 15 mM tyramine; *lgc-55(tm2813)*, *tdc-1(n3419)*, *ser-2(pk1357)*, and *tyra-3 (ok325)* mutants, 45 animals analyzed per tyramine concentration, three animals on each of 15 plates containing either water or 15 mM tyramine. B, 30 animals analyzed per tyramine concentration, three animals on each of 10 plates. C, 150 animals per genotype and transgene, three animals on each of 10 plates with or without tyramine from five separate transgenic lines. D-E, wild-type, 235 inactive and 157 active state intervals from 11 animals; *lgc-55(tm2913)*, 368 inactive and 101 active state intervals from nine animals.

## Acknowledgements

This work was funded by a postdoctoral fellowship from the American Heart Association to KMC (POST4990016) and by grants from the NINDS to MRK (NS036918) and to MRK and KMC (NS086932). Confocal instrumentation was supported by a grant to the Yale Liver Center (DK34989). Some strains were provided by the *C. elegans* Genetics Center, which is funded by NIH Office of Research Infrastructure Programs (P40 OD010440). We thank Navonil Banerjee and Mike Francis for sharing unpublished results. We are grateful to members of the Koelle and Collins labs, and James Baker, Julia Dallman, Mason Klein, and Sheyum Syed for comments and helpful discussions on the manuscript.

## Additional information

### Funding

| Funder | Grant reference number | Author |
|---|---|---|
| American Heart Association | Postdoctoral Fellowship, POST4990016 | Kevin M Collins |
| National Institute of Neurological Disorders and Stroke | NS086932 | Kevin M Collins Michael R Koelle |
| Yale Liver Center | DK34989 | Michael R Koelle |
| National Institute of Neurological Disorders and Stroke | NS036918 | Michael R Koelle |

The funders had no role in study design, data collection and interpretation, or the decision to submit the work for publication.

### Author contributions

KMC, Conception and design, Acquisition of data, Analysis and interpretation of data, Drafting or revising the article, Contributed unpublished essential data or reagents; AB, Conception and design, Acquisition of data, Analysis and interpretation of data, Drafting or revising the article; RWF, JET, Conception and design, Acquisition of data, Analysis and interpretation of data; JCB, Drafting or revising the article, Contributed unpublished essential data or reagents; MSC, Acquisition of data, Analysis and interpretation of data; MRK, Conception and design, Analysis and interpretation of data, Drafting or revising the article

### Author ORCIDs

Kevin M Collins, http://orcid.org/0000-0001-9930-0924
Jacob C Brewer, http://orcid.org/0000-0003-2780-2874
Michael R Koelle, http://orcid.org/0000-0001-9486-8481

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
