## [Decision Letter]

Thank you for submitting your article "Activity of the *C. elegans* egg-laying behavior circuit is controlled by competing activation and feedback inhibition" for consideration by *eLife*. Your article has been very favorably reviewed by two peer reviewers, including L René García (Reviewer #2), and the evaluation has been overseen by a Reviewing Editor, Oliver Hobert, and a Senior Editor.

You can see the reviewers’ comments below. We will gladly accept the manuscript pending some very minor editorial revisions suggested by the reviewers.

*Reviewer #1:*

Overall I think this is a very nice paper. Egg-laying is one of the best-described *C. elegans* behaviors, but the mechanistic details of how specific neurons and muscles control egg-laying have not been well-understood. The authors use optical imaging of neural and muscle cell activity in behaving worms to define the roles of specific motor neurons and accessory cells in generating egg-laying events. In particular they do a nice job relating their imaging data to previous work on the behavioral timing of egg-laying. The experiments are well-controlled and thorough, and justify the authors' conclusions. I am supportive of publication more or less as is.

One comment: in several places (for example subsection “Distinct patterns of activity in the egg-laying circuit accompany the active and inactive behavior states”, second paragraph) the authors refer to "rhythmic" activity in various cells in the egg-laying circuit. Although the traces seem like the activity could be rhythmic (i.e. events separated by more or less periodic intervals) they do not provide evidence for this. This is particularly relevant since some earlier behavioral work has described egg-laying events within the active phase as stochastic rather than periodic. It should be straightforward to see if the intervals between neuronal spikes have a regular period or alternatively follow some sort of random distribution; either outcome would be interesting.

Reviewer #2:

The article, "Activity of the *C. elegans* egg-laying behavior circuit is controlled by competing activation and feedback inhibition," primarily uses the fluorescence changes of G-CaMP5 to detect the temporal activities of cellular components used for egg-laying behavior. The theoretical foundations for how egg-laying behavior might be executed and regulated have been proposed in many earlier reports; however, this report uses G-CaMP-mediated real time calcium imaging of unrestrained behaving hermaphrodites to validate or re-qualify these ideas, and to generate more precise temporal measurements of activities between the HSN neurons, the VC neurons, the vulva muscles and the UV1 uterine endocrine cells.

The technical execution of measuring calcium- induced G-CaMP changes (relative to stable mCherry fluorescence) in moving hermaphrodites have been described in the authors' earlier 2013 report (my lab also does a lot of calcium imaging in *C. elegans*, albeit not in hermaphrodites, and generating this kind of data and analyzing them is not trivial). The methodology is sound and I do not have any concerns about how the experiments were conducted in this report.

In this report they use the G-CaMP imaging technology to support their idea that egg-laying behavior is coordinated with locomotion. The authors brought up this possibility in their 2013 J. Neuroscience paper, but in this study they measured several additional parameters to demonstrate that egg deposition has a higher probability of occurring when the ventral body wall muscles transitions between a relaxed and contracted state. Here they correlated the angles of worm postures with calcium transients of the HSN, VC neurons and vulva muscles with egg-laying events. What was surprising was the broad activity profiles of these excitable cells, but that only when the worm was in a specific posture (during locomotion), these cells activities correlated to an egg-laying event.

Although I know that correlation is not causation, I am excited (from a pure curiosity-loving standpoint) to see that kind of activity profile occurring in the hermaphrodite. The relationship between the specific phases in locomotion and the broad temporal activities in egg-laying components also suggested to the authors that ventral cord neurons (VA/VB) and gonadal cells are also supplying information to the HSN, VC neurons and vulva muscles. The authors followed up these interactions by perturbing other components of the egg-laying system.

Previous work has implicated that under certain laboratory conditions (such as laser-ablations, mutant backgrounds and pharmacological treatments), the HSN and VC neurons have some overlapping functions in regulating vulva muscle contractions. The authors here revisited some of these conditions using cell-targeted CH2R-activation, the *egl-1(n986dm)* allele and exploring tyramine signaling, to report a more detailed picture of the temporal activities of the HSN VC and vulva muscle changes relative to the locomotor and fertility state of the hermaphrodite. The work presented in these sections summarizes and clarifies some of the hypothesis made in earlier reports.

In summary, I enjoyed reading the report. I do not believe that there are any radical ideas or discoveries made in this report, but the report is a solid careful analysis of a complicated behavior. I also believe that the authors are deeply excited about understanding egg-laying behavior and very few other groups would put in the attention and detail to further our understanding of the problem. The approach of correlating HSN, VC UV1 and vulva muscle calcium activities with locomotion and egg-laying events is sound and validates hypotheses made in earlier reports by the authors and other in the field. The Discussion is well-written and summarizes the results, as-well-as raises a more mature way of thinking about how egg-laying behavior is executed in the hermaphrodite.

Minor Comments:

Figure 2 and Figure 4 took me awhile to understand, probably because these diagrams condense a lot of information in a presentation that is not standard. I was not clear how the data of individual hermaphrodites was parsed out or combined into this graph; that is how does the egg-laying behavior profiles differ between individual hermaphrodites.

I leave this up to the authors but maybe in a supplemental figure, the authors can present the data in a more traditional histogram presentation, so that the reader can see how the parameters obtained from individual hermaphrodites were processed to give the charts in Figure 2 and Figure 4.

---

## [Author Response]

*[…] Reviewer #1:*

*[…] One comment: in several places (for example subsection “Distinct patterns of activity in the egg-laying circuit accompany the active and inactive behavior states”, second paragraph) the authors refer to "rhythmic" activity in various cells in the egg-laying circuit. Although the traces seem like the activity could be rhythmic (i.e. events separated by more or less periodic intervals) they do not provide evidence for this. This is particularly relevant since some earlier behavioral work has described egg-laying events within the active phase as stochastic rather than periodic. It should be straightforward to see if the intervals between neuronal spikes have a regular period or alternatively follow some sort of random distribution; either outcome would be interesting.*

We have completed the analysis requested which is now part of an expanded Figure 2. We have conducted a power spectrum analysis of HSN, VC, and vulval muscle Ca^2+^ traces, and representative and summary results of this analysis are shown in Figure 2. We discuss our results finding ~50, ~100, and ~140 mHz activity rhythms found in the HSN, VC, and vulval muscles Ca^2+^ traces in the subsection “HSN, VC, and vulval muscle activity are rhythmic and phased with animal locomotion”, which then follows into the improved phasing presentation requested by Reviewer #2 (below). We outline in the Materials and methods how this analysis was performed in MATLAB (subsection “Ratiometric Ca^2+^ Imaging”, last paragraph).

Reviewer #2

*Minor Comments:*

*Figure 2 and Figure 4 took me awhile to understand, probably because these diagrams condense a lot of information in a presentation that is not standard. I was not clear how the data of individual hermaphrodites was parsed out or combined into this graph; that is how does the egg-laying behavior profiles differ between individual hermaphrodites.*

*I leave this up to the authors but maybe in a supplemental figure, the authors can present the data in a more traditional histogram presentation, so that the reader can see how the parameters obtained from individual hermaphrodites were processed to give the charts in Figure 2 and Figure 4.*

We agree that the circular histogram was not the easiest way to convey our observations of circuit activity being phased with locomotion. We have included additional information in the Materials and methods explaining how this analysis was conducted (subsection “Locomotion phasing analysis”). We have also included bar and whisker scatter plots of the primary phasing data separated by animal (Figure 2—figure supplement 1 and legend; and Figure 4—figure supplement 1 and legend). Finally, we have tried to improve the presentation using a horizontal line histogram that we hope will more easily communicate the results (Figure 2 and Figure 2—figure supplement 1).